# Coffin-Lowry Syndrome Induced by *RPS6KA3* Gene Variation in China: A Case Report in Twins

**DOI:** 10.3390/medicina58070958

**Published:** 2022-07-20

**Authors:** Huiying Jin, Haifeng Li, Shu Qiang

**Affiliations:** Department of Rehabilitation, The Children’s Hospital, Zhejiang University School of Medicine, National Clinical Research Center for Child Health, Hangzhou 310052, China; 6199005@zju.edu.cn

**Keywords:** twins, ribosomal protein S6 kinase polypeptide 3 (*RPS6KA3*), Coffin-Lowry Syndrome

## Abstract

Background and objectives: Coffin-Lowry Syndrome (CLS), a rare neurodegenerative disorder, is mainly diagnosed based on clinical manifestations and molecular analyses. In total, about 20 cases of CLS have been reported in China. Here, we report two cases of CLS in identical twin brothers and examine their potential causative mutations. Methods: The Trio mode was used in this analysis, i.e., DNA from the proband and his parents was sequenced. Furthermore, DNA from the proband’s twin brother was used for confirmation. Results: A hemizygous variation was detected in the 11th exon of the *RPS6KA3* gene, c.898C>T (p.R300*) of the proband, and the same site variation was detected in his identical twin brother; however, the mutation was not detected in his parents. Conclusions: The *RPS6KA3* gene mutation c.898C>T (p.R300*) is the causative factor of familial CLS. The variant detected was reported for the first time in the Chinese population. Additionally, by analyzing the previous literature, we were able to summarize the phenotypic and genetic characteristics of GLS in China.

## 1. Introduction

Coffin-Lowry Syndrome (CLS), a rare X-linked semi-dominant neurodegenerative genetic disorder, has a prevalence of 1:100,000–1:50,000 and occurs mostly in males [1]. The CLS pathogenic gene, ribosomal protein S6 kinase polypeptide 3 (*RPS6KA3*), is located at Xp22.2 [2]. More than 200 cases of CLS have been reported abroad, and 18 cases with confirmed pathogenic genes have been reported in China up to 2021 [3,4,5,6,7,8,9]. Here, we present the clinical data of two cases of CLS in twin brothers diagnosed and followed up at the Children’s Hospital affiliated with Zhejiang University School of Medicine (Hangzhou, China) in April 2022 and analyzed the phenotypic and genotypic characteristics of CLS cases in China.

## 2. Case Report

### 2.1. Clinical Data

The proband (child 1) was a male child who presented to the clinic at 5 months of age because head control was not achieved. He was born prematurely at 34 weeks with a birth weight of 1490 g. His Apgar score was 10 points. Physical examination showed characteristic face structures (wide eye distance, flat bridge of the nose), upright head instability, inability to raise his head in a prone position and low muscle tension in the limbs. The auxiliary examination included cranial magnetic resonance imaging (MRI), which revealed a widened external cerebral space, enlarged bilateral lateral ventricles and no obvious abnormal signal shadow in the brain parenchyma (see Figure 1). Color echocardiography revealed an enlarged coronary sinus and the persistent left superior vena cava (Figure 2). The auditory brainstem v-wave response threshold (air conductance) in the left ear was 30 dBnHL and that in the right ear was ≤15 dBnHL.

Child 2 was the identical twin of the proband. He had a birth weight of 1750 g and an Apgar score of 10 points. He was presented to the clinic along with the proband. Physical examination revealed similar results. Cranial MRI revealed an enlarged external cerebral space and bilateral lateral ventricles, high water content in the white matter of the bilateral frontal and parietal lobes and symmetrical dotted abnormal signals in bilateral frontal lobes. However, re-examination was recommended (see Figure 3). Color echocardiography revealed mild tricuspid regurgitation (Figure 4). The auditory brainstem v-wave response threshold (air conductance) in the left ear was 45 dBnHL and that in the right ear was 20 dBnHL.

The children were born from the first pregnancy of the parents. The father was 31 years old and the mother was 28 years old. The children were conceived naturally. Cytomegalovirus infection occurred during pregnancy. The twins were delivered via caesarean section because of a massive hemorrhage of the placenta previa at 34 weeks of gestation. There was no significant genetic history noted within three generations of the family.

### 2.2. Experimental Methods

#### 2.2.1. Test Sample Collection 

Peripheral blood (2 mL) was collected from the proband, child 2, and both parents for whole-exon sequencing. Genetic analysis was performed in the proband and site verification was performed in child 2. The study was approved by the Ethics Committee of the Children’s Hospital affiliated with Zhejiang University School of Medicine, and all samples were collected with written informed consent from the participants.

#### 2.2.2. Experimental Process

Genetic analysis was performed in the Children’s Hospital affiliated with Zhejiang University School of Medicine (Hangzhou, China). The test samples were used for genomic DNA extraction. DNA was then sheared and prepared for sequencing; the exons and adjacent intron regions of related genes were captured and enriched. The enriched target fragments were sequenced using a high-throughput sequencing platform. The data were interpreted based on the American College of Medical Genetics and Genomics guidelines. The variable was named based on the rules of the Human Genome Variation Society (http://www.hgvs.org/mutnomen/) (accessed on 14 April 2022).

### 2.3. Sequencing Results

A hemizygous variation was found in the *RPS6KA3* gene of the proband, in the 11th exon c.898C>T (p.R300*) (See Table 1 and Table 2 and Figure 5 and Figure 6). This variation is a nonsense mutation that can lead to the premature termination of polypeptide chain synthesis. This variation was also found in child 2 (Table 3).

## 3. Discussion

CLS is an X-linked dominant disorder characterized by moderate to severe intellectual disability, dystonia, craniofacial features, thinning of fingers, hearing impairment, short stature and skeletal deformity [7]. The diagnosis of CLS mainly depends on clinical manifestations and molecular analyses. The proband described in this report was admitted to our department at 5 months of age, and his main symptoms included specific facial features (wide eye distance and flat bridge of nose), motor retardation and dystonia. Cranial magnetic resonance imaging suggested the retardation of brain development and hearing impairment. The clinical manifestations of his twin brother were consistent, suggesting a high possibility of genetic disease.

Currently, more than 140 different inactivated variants are reported in patients with CLS, including 30% dislocation variants, 15% nonsense variants, 20% splicing errors and 30% microdeletions or insertions [1]. In this study, the *RPS6KA3* gene c.898C>T (p.R300*) was found to be a nonsense variant; two cases of CLS with a variation in this locus have been reported abroad [10,11]. Xiong et al. used machine learning technology to predict that this locus variation might affect normal mRNA splicing [12]. The present study represents the first report of this mutation in the Chinese population. According to the analysis of phenotype and gene mutation types of GLS reported in China (Table 4), the phenotype of the mutation at this locus has obvious special facial features, motor retardation and hypotonia, which should be given attention in clinical diagnosis.

With advancing molecular technology, molecular genetic techniques are being used for the diagnosis of genetic diseases, facilitating the early diagnosis and monitoring of patients. From the perspective of CLS cases, diagnosis in recent years has mostly relied on high-throughput sequencing technology [13,14,15]. In this study, a high-throughput sequencing platform was used for sequencing. The obtained DNA sequence was compared with the reference human genome sequence HG19 provided by the UCSC database, and the coverage and sequencing quality of the target region were evaluated. Bioinformatics and pathogenicity analyses were performed for variations with coverage of 20× and above.

Additional tests and follow up are important for patients diagnosed with CLS. These patients show progressive skeletal changes, including scoliosis and spondylolisthesis [16], which should be identified and treated during follow up. Cardiac abnormalities are also associated with CLS. For example, it is recommended that patients with CLS are screened for congenital structural heart defects and left ventricular noncompaction cardiomyopathy, a clinically heterogeneous disease [17]. Patients are reported to have recurrent episodes of nonconvulsive status epilepticus (NCSE) and status epilepticus. Among these, the diagnosis of NCSE might be missed [18] and this condition should be monitored during follow up.

## 4. Conclusions

Overall, this study provides evidence for the diagnosis of CLS in twin children based on consistent special facial features, motor retardation and low muscle tone. Additionally, the *RPS6KA3* gene mutation c.898C>T (p.R300*), detected by whole-exon sequencing in this study, was reported for the first time in the Chinese population. Follow-up treatment and follow up are recommended to focus on skeletal deformities, epilepsy and cardiomyopathy.

## Figures and Tables

**Figure 1 medicina-58-00958-f001:**
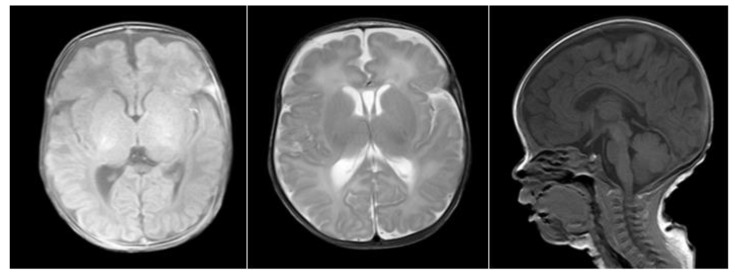
Proband’s cranial MRI (TIW−SE, T2W−SE, TIW−SAG).

**Figure 2 medicina-58-00958-f002:**
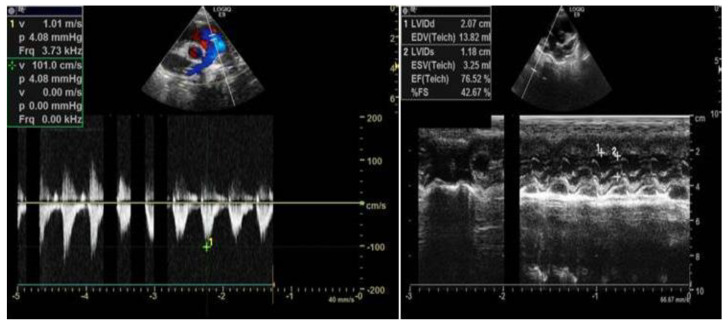
Proband’s heart color ultrasound.

**Figure 3 medicina-58-00958-f003:**
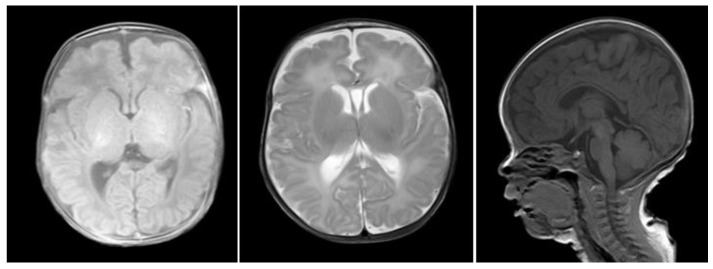
Child 2’s cranial MRI (TIW−SE, T2W−SE, TIW−SAG).

**Figure 4 medicina-58-00958-f004:**
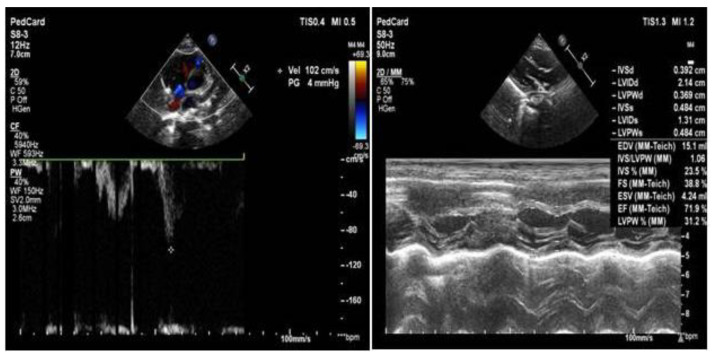
Child 2’s heart color ultrasound.

**Figure 5 medicina-58-00958-f005:**
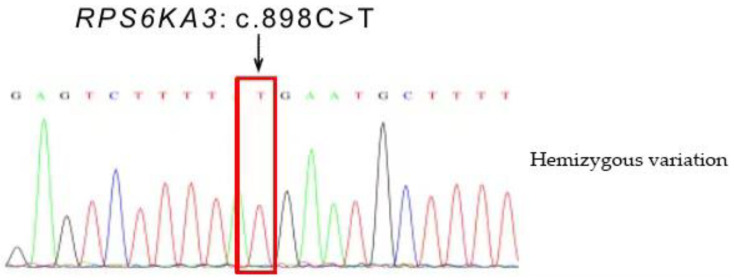
Distribution of signal intensity for each chromosome (proband/father/mother).

**Figure 6 medicina-58-00958-f006:**
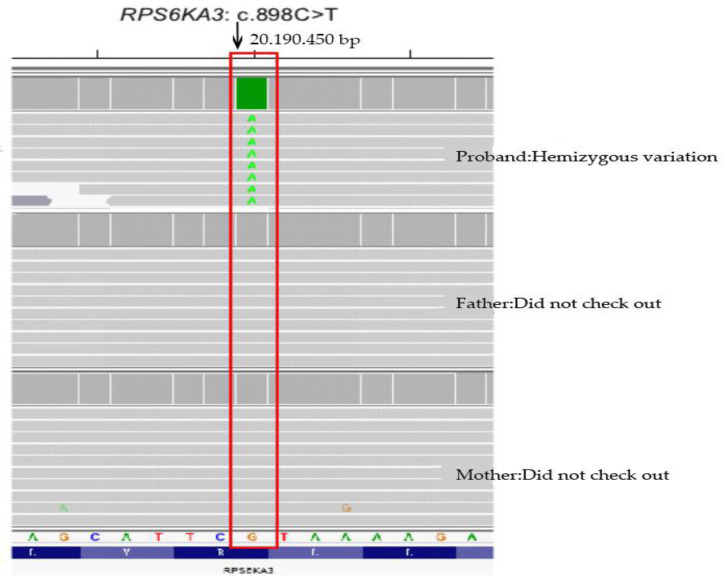
Integrative Genomics Viewer (IGV) diagram of second-generation sequencing (proband/father/mother) Note: *RPS6KA3* gene is reverse encoding.

**Table 1 medicina-58-00958-t001:** Proband sequencing results.

Gene	Chromosome Position	Transcript Number	Exon/Intron	Nucleotide Change	Amino Acid Change	Heterozygous/ Homozygous	Related Diseases	Inheritance Mode	Variation Classification	Source of Variation
*RPS6KA3*	chrX: 20195150	NM_004586.3	Exon 11	c.898C>T	p.R300*	Hemizygous	Coffin-Lowry syndrome	XLD and XLD	Pathogenic	De novo mutation

Note: The reference gene version was GRCh37/HG19 and XLD was X-linked dominant.

**Table 2 medicina-58-00958-t002:** Sequencing quality statistics of the target region.

The Sample Name	Sample Number	Test Data Volume (bp)	Mean Sequencing Depth	Target Area Coverage	Proportion of Coverage Area over 10×	Proportion of Coverage Area over 20×
Proband	WES22030470	17376448800	220.48	99.97%	99.90%	99.77%
Father	WES22030471	19205799600	244.88	99.96%	99.90%	99.78%
Mother	WES22030472	14236229100	181.27	99.82%	99.69%	99.48%

**Table 3 medicina-58-00958-t003:** Child 2 sequencing result.

Gene	Chromosome Position	Transcript Number	Exon/Intron	Nucleotide Change	Amino Acid Change	Heterozygous/ Homozygous	Variation Classification	Source of Variation
*RPS6KA3*	chrX: 20195150	NM_004586.3	Exon 11	c.898C>T	p.R300*	Hemizygous	Pathogenic	De novo mutation

**Table 4 medicina-58-00958-t004:** The phenotypic and genetic characteristics of GLS in China.

	Case	Age of Diagnosis	Gender	Special Facial Features	Tapered Fingers	Developmental Delay	Hypotonia	Abnormal Hearing	Cranial MRI Abnormalities	*RPS6KA3* Mutation
This study	1	5M	M	+	-	+	+	+	+	Exon 11:c.898C>T (p.R300*)
	2	5M	M	+	-	+	+	+	+	Exon 11:c.898C>T (p.R300*)
Li, Y [3]	3	2Y6M	M	+	+	+	+	N/A	+	N/A
Wang, Y [4]	4	13Y	M	+	+	+	+	N/A	N/A	c.r889_890delAG
Zhang, L [5]	5	4Y3M	M	+	+	+	+	-	-	Exon 5:c.340C>T
Shen, N [6]	6	2Y	M	+	+	+	+	+	-	Exon12:c.966_967delAA(p.Arg323Thr fs*11)
Liu, Y [7]	7	1Y	M	+	-	+	+	+	N/A	C.1672cC> T (p.R558*)
	8	4Y	M	+	-	+	+	+	+	Hemizygote variation(c.325+2_325+3insT)
Li, Q [8]	9	3Y4M	M	+	+	+	+	-	-	Exon 19:c.1841+1G>A
Fung, J.L [9]	10	10Y	M	+	+	+	+	-	N/A	Deletion of exon 9 and 10
	11	36Y	F	+	+	N/A	N/A	-	N/A	Deletion of exon 9 and 10
	12	3Y	M	+	+	+	+	+	N/A	Exon17: c.1449T>A, p.(Tyr483Ter)
	13	19Y	F	+	+	+	+	-	N/A	Exon 17: c.1449T>A, p.(Tyr483Ter
	14	42Y	F	+	+	+	N/A	-	N/A	Exon 17: c.1449T>A, p.(Tyr483Ter)
	15	2Y	M	+	+	+	+	-	N/A	c.1842-1G>T
	16	25Y	M	+	+	+	+	-	N/A	Exon 16: c.1428_ 1430delTAT
	17	18Y	M	+	+	+	+	-	N/A	Exon 9: c.638G>A, p.(Gly213Asp)
	18	16Y	F	+	+	+	-	-	N/A	Exon 7: c.501delA, p.(Glu168fsTer14)

Abbreviations: +, positive: -, negative; N/A, not available.

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
