# Peer review of "Coffin-Lowry Syndrome Induced by RPS6KA3 Gene Variation in China: A Case Report in Twins"

_medicina, 2022, doi:10.3390/medicina58070958_

Round 1

Reviewer 1 Report

some grammatical errors need correction

Author Response

We really appreciate you for your carefulness and conscientiousness. Your suggestions are really valuable and helpful for revising and improving our paper. According to your suggestions, we have made the following revisions on this manuscript:  

some grammatical errors need correction

 Response:Thank you very much for your advice. We have modified it.

Reviewer 2 Report

Overall the case report is rather descriptive novel variants resulting from novel phenotypes adds more value hence I suggest to do a through clinical ( phenotypic match ) evaluation to find if there are any novel features that cannot be attributed to the CMV infection in the mother. Otherwise you need to do some functional studies. 

The amino acid change should be uniform e.g - (P.r300 *) should be ( p. R300 *) 

Introduction should be expanded if there is no stipulated word limit 

Many grammatical errors 

Re- phrase “because he could not raise his head” to since head control was not achieved 

“characteristic face structure” should be characteristic facial structures 

Child 2 - should be lower case unless its in the start of a sentence 

Discussion 

Were there any features unique in this case which might have resulted from the novel variant ? 

I would suggest to add a table with reported phenotypes for CLS and match them to your present case  

Have you ruled out features that could be due to CMV infection in the mother ? Few lines about this will benefit the paper 

Author Response

We really appreciate you for your carefulness and conscientiousness. Your suggestions are really valuable and helpful for revising and improving our paper. According to your suggestions, we have made the following revisions on this manuscript:  

Overall the case report is rather descriptive novel variants resulting from novel phenotypes adds more value hence I suggest to do a through clinical ( phenotypic match ) evaluation to find if there are any novel features that cannot be attributed to the CMV infection in the mother. Otherwise you need to do some functional studies.

Response:Thank you very much for your advice. We have modified it.The proband physical examination showed characteristic face structures (wide eye distance, flat bridge of the nose),which are different from CMV infection((page 2, lines 39-40)).We added Table 4 to summarize the phenotypes in this case, (page 8, lines 134).

The amino acid change should be uniform e.g - (P.r300 *) should be ( p. R300 *)

Introduction should be expanded if there is no stipulated word limit

Response:Thank you very much for your advice. We have modified it(page 1, lines 15).

We have expanded Introduction(page 1, lines 30-33).

Many grammatical errors

Re- phrase “because he could not raise his head” to since head control was not achieved

“characteristic face structure” should be characteristic facial structures

Child 2 - should be lower case unless its in the start of a sentence

Response:Thank you very much for your advice. We have modified it(page 2, lines 37、39).

Discussion

Were there any features unique in this case which might have resulted from the novel variant ?

Response:Thank you very much for your advice. We have modified it.This case was diagnosed at the age of 5 months, which is the youngest among the reported cases in China.the phenotype of the mutation at this locus has obvious special facial features, motor retardation and hypotonia, which should be paid attention to in clinical diagnosis(page 7, lines 130-134).

I would suggest to add a table with reported phenotypes for CLS and match them to your present case  

Response:Thank you very much for your advice. We have added it(Table 4,page 8, lines 135).

Have you ruled out features that could be due to CMV infection in the mother ? Few lines about this will benefit the paper

Response:Thank you very much for your advice. We added Table 4 to summarize the phenotype and genotype of this case, which is different from CMV infection(Table 4,page 8, lines 135).

Reviewer 3 Report

1) The authors can improve the discussion with a more interesting background. Moreover, they should specify the objective of this report in the introduction.

2) Could the authors provide images of the dysmorphic facial features?

3) Did the parents or any relative have any similar morphologic similarity with Coffin-Lowry Syndrome?

4) Could the CMV infection that occurred during pregnancy be related to the outcomes in the twins?

5) Who did the experimental process of genomic DNA extraction?

6) It is advised to provide a table with ‘‘near’’ mutations and the relationship with clinical manifestations. Moreover, the authors could present a table showing the variation of symptoms based on RPS6KA3 gene mutation local.

7) The authors should provide a table with other cases reported in the literature. This would greatly impact the quality of the manuscript.

Author Response

We really appreciate you for your carefulness and conscientiousness. Your suggestions are really valuable and helpful for revising and improving our paper. According to your suggestions, we have made the following revisions on this manuscript:  

1)The authors can improve the discussion with a more interesting background. Moreover, they should specify the objective of this report in the introduction.

Response:Thank you very much for your advice. We have modified it(page 1, lines 30-34).

2) Could the authors provide images of the dysmorphic facial features?

Response:Thank you very much for your advice.We consulted the parents, who did not agree to publish the mugshot

3) Did the parents or any relative have any similar morphologic similarity with Coffin-Lowry Syndrome?

Response:Thank you very much for your advice.There was no significant genetic history was noted within three generations of the family.We added it(page 4, lines 73-74).

4) Could the CMV infection that occurred during pregnancy be related to the outcomes in the twins?

Response:Thank you very much for your advice.We searched the literature, but found no evidence that CMV infection was related to GLS cases. There were only 2 cases in the medical record this time, so it is difficult to prove that there is a connection between the two cases. More cases may be needed to know.

5) Who did the experimental process of genomic DNA extraction?

Response:Thank you very much for your advicem,we added it.Experimental process of genomic DNA extraction was conducted in Children's Hospital affiliated to Zhejiang University School of Medicine (Hangzhou, China)page 4, lines84-85).

6) It is advised to provide a table with ‘‘near’’ mutations and the relationship with clinical manifestations. Moreover, the authors could present a table showing the variation of symptoms based on RPS6KA3 gene mutation local.

Response:Thank you very much for your advice.We have added it(Table 4,page 8, lines 135).

7)The authors should provide a table with other cases reported in the literature. This would greatly impact the quality of the manuscript.

Response:Thank you very much for your advice. We added an analysis of cases cited in the literature(Table 4,page 8, lines 135).

Round 2

Reviewer 2 Report

Yes I’m ok with the changes please proceed.

Reviewer 3 Report

Satisfactory